# Thromboembolic Events after COVID-19 Vaccination: An Italian Retrospective Real-World Safety Study

**DOI:** 10.3390/vaccines11101575

**Published:** 2023-10-10

**Authors:** Francesca Futura Bernardi, Annamaria Mascolo, Marina Sarno, Nicolina Capoluongo, Ugo Trama, Rosanna Ruggiero, Liberata Sportiello, Giovanni Maria Fusco, Massimo Bisogno, Enrico Coscioni, Anna Iervolino, Pierpaolo Di Micco, Annalisa Capuano, Alessandro Perrella

**Affiliations:** 1Directorate-General for Health Protection, Campania Region, 80143 Naples, Italy; bernardi.francesca.futura@gmail.com (F.F.B.); ugo.trama@regione.campania.it (U.T.); giom.fusco@gmail.com (G.M.F.); 2Campania Regional Centre for Pharmacovigilance and Pharmacoepidemiology, 80138 Naples, Italy; rosanna.ruggiero@unicampania.it (R.R.); liberata.sportiello@unicampania.it (L.S.); annalisa.capuano@unicampania.it (A.C.); 3Department of Experimental Medicine, Section of Pharmacology “L. Donatelli”, University of Campania “Luigi Vanvitelli”, Via Costantinopoli 16, 80138 Naples, Italy; 4Unit Emerging Infectious Disease, Ospedali dei Colli, P.O. D. Cotugno, 80131 Naples, Italy; marinasarno92@gmail.com (M.S.); nicolina.capoluongo@ospedalideicolli.it (N.C.); alessandro.perrella@ospedalideicolli.it (A.P.); 5Regional Special Office for Digital Transformation, Campania Region, 80100 Naples, Italy; massimo.bisogno@regione.campania.it; 6Division of Cardiac Surgery, AOU San Giovanni di Dio e Ruggi d’Aragona, 84131 Salerno, Italy; coscionienrico@gmail.com; 7Directorate-General AORN Ospedali dei Colli, Campania Region, 80131 Naples, Italy; anna.iervolino@ospedalideicolli.it; 8General Medicine, Santa Maria delle Grazie Hospital, ASL NA2 Nord, 80078 Pozzuoli, Italy; pdimicco@libero.it; 9Regional Observatory for Infectious Disease, Campania Region, 80131 Naples, Italy

**Keywords:** COVID-19, vaccination, thromboembolic events, safety

## Abstract

Introduction: Real-world safety studies can provide important evidence on the thromboembolic risk associated with COVID-19 vaccines, considering that millions of people have been already vaccinated against COVID-19. In this study, we aimed to estimate the incidence of thromboembolic events after COVID-19 vaccination and to compare the Oxford–AstraZeneca vaccine with other COVID-19 vaccines. Methods: We conducted a retrospective real-world safety study using data from two different data sources: the Italian Pharmacovigilance database (Rete Nazionale di Farmacovigilanza, RNF) and the Campania Region Health system (Sistema INFOrmativo saNità CampanIA, SINFONIA). From the start date of the COVID-19 vaccination campaign (27 December 2021) to 27 September 2022, information on COVID-19 vaccinations and thromboembolic events were extracted from the two databases. The reporting rate (RR) and its 95% confidence interval (95%CI) of thromboembolic events for 10,000 doses was calculated for each COVID-19 vaccine. Moreover, the odds of being vaccinated with the Oxford–AstraZeneca vaccine vs. the other COVID-19 vaccines in cases with thromboembolic events vs. controls without thromboembolic events were computed. Results: A total of 12,692,852 vaccine doses were administered in the Campania Region, of which 6,509,475 (51.28%) were in females and mostly related to the Pfizer-BioNtech vaccine (65.05%), followed by Moderna (24.31%), Oxford–AstraZeneca (9.71%), Janssen (0.91%), and Novavax (0.02%) vaccines. A total of 641 ICSRs with COVID-19 vaccines and vascular events were retrieved from the RNF for the Campania Region, of which 453 (70.67%) were in females. Most ICSRs reported the Pfizer-BioNtech vaccine (65.05%), followed by Oxford–AstraZeneca (9.71%), Moderna (24.31%), and Janssen (0.91%). A total of 2451 events were reported in the ICSRs (3.8 events for ICSRs), of which 292 were thromboembolic events. The higher RRs of thromboembolic events were found with the Oxford–AstraZeneca vaccine (RR: 4.62, 95%CI: 3.50–5.99) and Janssen vaccine (RR: 3.45, 95%CI: 0.94–8.82). Thromboembolic events were associated with a higher likelihood of exposure to the Oxford–AstraZeneca vaccine compared to Pfizer-BioNtech (OR: 6.06; 95%CI: 4.22–8.68) and Moderna vaccines (OR: 6.46; 95%CI: 4.00–10.80). Conclusion: We observed a higher reporting of thromboembolic events with viral-vector-based vaccines (Oxford–AstraZeneca and Janssen) and an increased likelihood of being exposed to the Oxford–AstraZeneca vaccine compared to the mRNA vaccines (Pfizer-BioNtech and Moderna) among thromboembolic cases.

## 1. Introduction

The rapid development of coronavirus disease 2019 (COVID-19) vaccines has helped change the course of the pandemic by reducing illness severity and hospital admissions [1]. As of January 2023, the European Medicines Agency (EMA) had granted authorization to seven COVID-19 vaccines [2]. Among these, two are mRNA vaccines: BNT162b2 mRNA (produced by Pfizer-BioNTech, approved on 21 December 2020) and mRNA-1273 (Moderna, approved on 6 January 2021). Additionally, two vaccines are based on adenovirus technology: ChAdOx1 nCoV-19 (Oxford–AstraZeneca, approved on 29 January 2021) and Ad.26.COV2.S (Janssen, approved on 11 March 2021); two of them are two recombinant-protein-based adjuvanted vaccines: NVX-CoV2373 (Novavax, 20 December 201) and Vidprevtyn (Sanofi, 10 November 2022); and one is an inactivated adjuvanted vaccine: VLA2001 (Valneva, 24 June 2022). Since their introduction, these vaccines have been associated with safety concerns [3]. Specifically, in March 2021, in response to spontaneous reports of thromboembolic events coupled with thrombocytopenia among individuals who received the Oxford–AstraZeneca vaccine, several European countries temporarily halted the administration of this vaccine [4]. Initially, these reports included 62 cases of cerebral venous sinus thrombosis (CVST) and 24 cases of splanchnic venous thrombosis (SVT) were reported with the Oxford–AstraZeneca vaccine in the European Union and United Kingdom (with 25,000,000 doses administered) [5]. In April 2021, thromboembolic events with thrombocytopenia were also reported with the Janssen vaccine [6]. The Pharmacovigilance Risk Assessment Committee (PRAC) of the EMA confirmed a plausible causal relationship between rare events of thrombosis with thrombocytopenia and adenovirus-based vaccines [7,8]. Subsequently, in May 2021, the EMA published recommendations for the monitoring and prevention of thrombosis with the Oxford–AstraZeneca vaccine [9]; in November 2021, cases of CVST without thrombocytopenia were also observed with this vaccine leading the PRAC to amend its Summary of Product Characteristics to include this adverse event [10]. In the literature, two population-based studies showed an increased risk of thromboembolic events among people vaccinated with one dose of the Oxford–AstraZeneca vaccine compared to that observed in the general population [11,12]. Although far less reported, such events were also observed with mRNA vaccines [13]. A population-based studies conducted on over 2 million people vaccinated against COVID-19 found a risk of pulmonary embolism with the Pfizer-BioNtech vaccine and a risk of thrombocytopenia following the Pfizer-BioNtech and Oxford–AstraZeneca vaccinations [3]. In this context, real-world safety studies can provide important evidence on the safety of COVID-19 vaccines, considering that millions of people have been already vaccinated against COVID-19. In this study, we aimed to estimate the incidence of thromboembolic events after COVID-19 vaccination and to compare the Oxford–AstraZeneca vaccine with other COVID-19 vaccines using real-life data.

## 2. Methods

### 2.1. Study Design and Data Source

We conducted a retrospective real-world safety study using data from two different data sources: the Italian Pharmacovigilance database (Rete Nazionale di Farmacovigilanza, RNF) and the Campania Region Health system (Sistema INFOrmativo saNità CampanIA, SINFONIA). The RNF is the pharmacovigilance database managed by the Italian Medicine Agency (Agenzia Italiana del Farmaco, AIFA) containing all spontaneous reports of suspected adverse reactions to medicinal products or vaccines relevant to the national territory. The spontaneous reports were recorded according to general practitioners, physician evaluations, or any other healthcare professional. SINFONIA is the regional health information system designed to support the entire health service of the Campania Region (South of Italy) by increasing efficiency, containing costs, and enhancing the needs of all healthcare players (healthcare professionals, citizens, health structures, and institutions). SINFONIA contains socio-demographic and healthcare data of all Campania Region residents, including data on COVID-19 vaccinations.

### 2.2. Data Extraction

For the period from the start date of the COVID-19 vaccination campaign (27 December 2021) to 27 September 2022, information about COVID-19 vaccinations administered in the Campania Region, including patients’ characteristics (age and sex), type of vaccine, and vaccine dose, were extracted from SINFONIA in an anonymized and aggregated form according to our previous research protocol [14]. Briefly Machine learning (ML) analysis was conducted using a Python scripting model within the Spyder IDE (64-bit version). The analysis aimed to select, extract, and match individuals based on their vaccination status (vaccinated/negative or vaccinated/positive) and to perform forecasting analysis on contagiousness and coverage trends. The obtained data were further organized using Tableau Professional. In essence, machine learning algorithms leverage data to establish relationships between different data points and then export this information to another software for statistical analysis and graphical representation. Additionally, these algorithms create predictive models to forecast future trends and outcomes. These models essentially represent the actions the machine will take to achieve a specific result. This approach takes into account three primary elements: Event (positive or negative test), Category (type and time of subject—vaccinated or unvaccinated), and Time (t). The following formula is used for trend evaluation: Time series (Ts) = (Ep/En + Cv/Cu). During the same period, Individual Case Safety Reports (ICSRs) reporting at least one COVID-19 vaccine as suspected and one vascular event were retrieved from the RNF for the Campania Region. Vascular events were recognized using the System Organ Class (SOC) “Vascular disorders” of the Medical Dictionary for Regulatory Activities (MedDRA), version 25.1. From ICSRs, the information on patients’ characteristics (age and sex), thromboembolic events, and seriousness was available. In accordance with the International Council on Harmonization E2D guidelines, a case was serious if at least one adverse event was life-threatening, resulted in death, caused/prolonged hospitalization, was disabling, determined a congenital anomaly/birth defect, or was another medically important condition. On 27 December 2021, the following COVID-19 vaccines were administered in the Campania Region: Pfizer-BioNtech, Moderna, Oxford–AstraZeneca, Janssen, and Novavax vaccines. Therefore, data were retrieved only for those aforementioned vaccines. The adverse drug reaction severity classification has been made according to the FDA.

### 2.3. Descriptive and Statistical Analyses

Descriptive analyses were performed separately for the information reported in the two data sources and illustrated as numbers and percentages. The reporting rate (RR) and its 95% confidence interval (95%CI) of thromboembolic events for 10,000 doses was calculated for each COVID-19 vaccine by dividing the numbers of events with the number of doses administered. For the association between thromboembolic events and COVID-19 vaccinations, the odds of being vaccinated with the Oxford–AstraZeneca vaccine vs. the other COVID-19 vaccines in cases with thromboembolic events vs. controls without thromboembolic events were computed. The odds ratios (ORs) and their 95%CI were applied for all comparisons. A 5% significance level was considered for analyses that were performed with R (version 3.2.2, R Development Core Team). 

### 2.4. Ethics

According to the local legislation, a retrospective pharmacovigilance study does not require ethical approval. For the use of retrospective SINFONIA data, this study was conducted in accordance with the Declaration of Helsinki 1975 and its later amendments. The research did not involve a clinical study, and all patients’ data were fully anonymized and were analyzed retrospectively. For this type of study, formal consent was not required according to the current national established by the Italian Medicines Agency, and according to the Italian Data Protection Authority, neither ethical committee approval nor informed consent were required for anonymized data, as confirmed and approved by the Ethical Committee of “Aziende Ospedaliere di Rilievo Nazionale e di Alta Specializzazione—A.Cardarelli/Santobono—Pausilipon” (Protocol Number 00000926 of 11 January 2022).

## 3. Results

### 3.1. Descriptive Results from SINFONIA

During the study period, a total of 12,692,852 vaccine doses were administered in the Campania Region, of which 6,509,475 (51.28%) were in females. Most doses were administered in patients aged 12–69 years (N = 10,535,109; 91.56%), with the age group of 50–59 years as the most representative (N = 2,232,236; 17.59%). Most doses were related to the Pfizer-BioNtech vaccine (65.05%), followed by Moderna (24.31%), Oxford–AstraZeneca (9.71%), Janssen (0.91%), and Novavax (0.02%) vaccines. All data from SINFONIA are displayed in Table 1.

### 3.2. Descriptive Results from RNF

During the study period, a total of 641 ICSRs with COVID-19 vaccines and vascular events were retrieved from the RNF for the Campania Region, of which 453 (70.67%) were females aged over 40 years old. Most ICSRs reported the Pfizer-BioNtech vaccine (65.05%), followed by Oxford–AstraZeneca (9.71%), Moderna (24.31%), and Janssen (0.91%). One case reported both Oxford–AstraZeneca and Moderna vaccines as suspected vaccines. No case of vascular events related to Novavax vaccine was found. The majority of ICSRs referred to the first dose of COVID-19 vaccines (N = 329; 51.32%), followed by second (N = 159; 24.81%) and third doses (N = 52; 8.11%). The information on doses was not available for 101 ICSRs (15.76%). Most ICSRs reported at least one event classified as serious (N = 261; 40.72%). Characteristics of ICSRs for each COVID-19 vaccine are reported in Table 2. A total of 2451 events were observed in the ICSRs (3.8 events for ICSRs), of which 292 were thromboembolic events. The most reported event was thrombosis (N = 17), followed by venous thrombosis (N = 10), deep venous thrombosis (N = 9), and D-dimer of fibrin increased (N = 9). Thromboembolic events for each COVID-19 vaccine are illustrated in Table 3, while all events are in Appendix A.

### 3.3. Statistical Results

The highest RRs of thromboembolic events were found with the Oxford–AstraZeneca vaccine (RR: 4.62, 95%CI: 3.50–5.99) and Janssen vaccine (RR: 3.45, 95%CI: 0.94–8.82). All RRs are reported in Table 4. Thromboembolic events were associated with a higher likelihood of exposure to the Oxford–AstraZeneca vaccine compared to Pfizer-BioNtech (OR: 6.06; 95%CI: 4.22–8.68) and Moderna vaccines (OR: 6.46; 95%CI: 4.00–10.80). No difference was instead observed in the comparison with the Janssen vaccine (Figure 1).

## 4. Discussion

This study involving 12,692,852 vaccine doses administered in the Campania Region, Italy, showed increased RRs of thromboembolic events with the viral-vector-based vaccines (Oxford–AstraZeneca and Janssen). Moreover, an increased likelihood of being exposed to the Oxford–AstraZeneca vaccine compared to Pfizer-BioNtech and Moderna vaccines among thromboembolic cases was found. For instance, the ORs were 506- and 546-fold for thromboembolic events, respectively. These findings are consistent with the thromboembolic risk observed in clinical studies and issued by EMA [7,9,10,15]. 

In the literature, clinical studies on thromboembolic events and COVID-19 vaccinations are limited and with controversial results according to different studied population. A study conducted on VAERS data found no significantly increased risk after mRNA vaccines [16], and a Danish retrospective cohort also showed no statistically significant association between the onset of thromboembolic or thrombocytopenic events and the Pfizer-BioNtech vaccine [16]. Moreover, a real-world evidence-based study conducted on patients in the Mayo Clinic Health System observed that CVST is a rare event not significantly associated with COVID-19 vaccines [17]. One international network cohort study compared the thrombotic risk between COVID-19 vaccines. Similarly to our findings, this study showed an increased risk of thrombocytopenia when the Oxford–AstraZeneca vaccine was compared to Pfizer-BioNtech vaccine (pooled calibrated incidence rate ratio 1.33; 95%CI: 1.18–1.50) [18]. A Danish and Norwegian cohort study on 281,264 recipients aged 18–65 years found an increased standardized incidence rate of venous thromboembolism (1.97; 95%CI: 1.50–2.54) and thrombocytopenia (3.02; 95%CI: 1.76–4.83) within 28 days of vaccination [11]. Rates of venous thromboembolism were largely driven by events of CVST. Further, this study did not observe any increased rates of arterial thromboembolism [11]. A nested Scottish case–control study found increased rates of idiopathic thrombocytopenic purpura (ITP), arterial thromboembolism, and hemorrhagic events in 1.7 million recipients of a first dose of the Oxford–AstraZeneca vaccine [12]. A similar risk of ITP was also found in a post hoc self-controlled case series analysis [13]. Another similar study showed an increased risk of hospital admissions or deaths due to events of thromboembolism among vaccine recipients. Specifically, the first dose of the Oxford–AstraZeneca vaccine was associated with thrombocytopenia, venous thromboembolism, and CVST, while the first dose of the Pfizer-BioNtech vaccine was associated with arterial thromboembolism, ischaemic stroke, and CVST [19]. In contrast, a cohort study did not find an overall association between Pfizer-BioNtech and arterial thromboembolism, but it was found in a subgroup analysis of recipients aged 50–69 years [3]. Another cohort study also showed an increased risk of pulmonary embolism among recipients of a first dose of the Pfizer-BioNtech vaccine [20]. Moreover, risks of pulmonary embolism and thrombocytopenia were found to be increased in recipients of the Oxford–AstraZeneca vaccine, and the risk of immune thrombocytopenia was found to be high for both Oxford–AstraZeneca and Pfizer-BioNtech vaccines [20]. In our study, due to the limited number of each specific event, we could only perform an analysis for the aggregated number of thromboembolic events. However, in terms of reporting, we observed that venous thromboembolism events were more reported than arterial thromboembolism events and reported after the first dose of vaccination, also in accordance with results from a systematic review [14].

The connection between the potential risk of hemorrhagic stroke and BNT162b2 remains unclear. This uncertainty could arise from the interaction between the spike protein of SARS-CoV-2 and platelets, potentially elevating the risk of thromboembolic events in SARS-CoV-2-infected patients, thereby contributing to significant bleeding incidents. The spike protein, targeted by both mRNA- and vector-based vaccines, might lead to a syndrome resembling thrombosis and thrombocytopenia in vaccine recipients, similar to heparin-induced thrombocytopenia in patients [21]. This parallels the current understanding of the observed risk of thrombocytopenic thrombosis associated with Vaxzevria [19]. Despite this, the mechanisms responsible for a potential link between COVID-19 vaccines and thromboembolic events are currently under investigation. Following the initial alerts regarding the Oxford–AstraZeneca vaccine, a new immune disorder known as vaccine-induced immune thrombotic thrombocytopenia emerged is being described in the literature for this vaccine [21,22]. This new event occurs as an atypical thrombosis associated with thrombocytopenia, including CVST, from 5 to 15 days after the vaccination, and it might be mediated by the cross-reactivity between antibodies generated after vaccination and platelet factor 4 (PF4) [14]. However, there is also evidence that does not support this hypothesis [23]. Some authors have also proposed that the inflammatory response after vaccination may increase the clearance mediated by macrophages and/or reduce the platelet production, thus inducing thrombocytopenia [13]. These mechanisms have been previously postulated for the ITP following viral infections [24], as well as vaccination against other viruses (such as measles-mumps-rubella and varicella-zoster) [25,26]. Another mechanism specific for viral-vector-based vaccines involves the adenovirus carrier deliveries of DNA encoding the Spike (S) protein to the pulmonary megakaryocytes via the coxsackie-adenovirus receptor (CAR). This leads to megakaryocyte activation, biogenesis of activated platelets, and release of thromboxane A2 (TxA2) and PF4 that further activates platelets and their traversal through the cerebral vein sinuses, leading to thromboinflammation, CVST, and thromboembolism in other blood vessels [27,28,29]. Moreover, vaccines containing the adenovirus as vector may bind the PF4, leading to the formation of an immunogenic complex (like the heparin–PF4 complex), which can cause platelet activation and thrombosis [30]. Generally, females are associated with a more pronounced platelet activation, and hence, a higher risk of thromboembolic events after vaccination [31]. Indeed, a higher reporting of thromboembolic events in females (71%) was observed.

To the best of our knowledge, the present study is the first conducted on BigData from a unique Italian Region (Campania) according to ML and AI algorithms [14]. The strength is the collection of aggregated data for more than 12,000,000 people vaccinated for COVID-19 using the large regional SINFONIA database that, as previously observed for other studies, represented the strength of our current research protocols [14,32]. Nonetheless, one of the significant limitations is the use of safety data from the spontaneous reporting system (the RNF), which is characterized by underreporting and poor quality of information. The underreporting could limit the observation of the real number of thromboembolic events that occurred in our regional territory. However, considering that a lower number of events may be identified, we may have underestimated rates of thrombotic adverse events. The poor quality of information limits our analysis from considering risk factors for thromboembolic events as well as the time between the vaccination and the onset of thromboembolic events. Furthermore, we could only use aggregated data from the SINFONIA database, therefore no information on vaccine doses for each COVID-19 vaccine was retrievable, and the analyses were limited on the total number of vaccine doses administered. Finally, an important limitation is that events retrieved from pharmacovigilance cases are not surely related to the vaccine but simply reported as a clinically significant event after vaccination. This latter concept is important to be highlighted in the pharmacovigilance field, and in fact, the spontaneous reporting is driven by the suspect and not even the certainty of an event–vaccine association.

## 5. Conclusions

In conclusion, we found a higher reporting of reported thromboembolic events with viral-vector-based vaccines (Oxford–AstraZeneca and Janssen) and an increased likelihood of being exposed to the Oxford–AstraZeneca vaccine compared to the mRNA vaccines (Pfizer-BioNtech and Moderna) among thromboembolic cases. COVID-19 vaccination remains the most effective prevention strategy to fight this pandemic, and any safety concern should be weighed against the advantages of being vaccinated. The continuous pharmacovigilance monitoring is fundamental to collect more information and help to improve the management of these rare but often severe thromboembolic events associated with COVID-19 vaccination. Further, the use of BigData, as SINFONIA in the Campania Region, associated with a machine learning algorithm for data extraction and analysis granted the possibility to have a deeper evaluation and potentially more helpful support in the evaluation of adverse events to drugs including vaccines [33].

## Figures and Tables

**Figure 1 vaccines-11-01575-f001:**
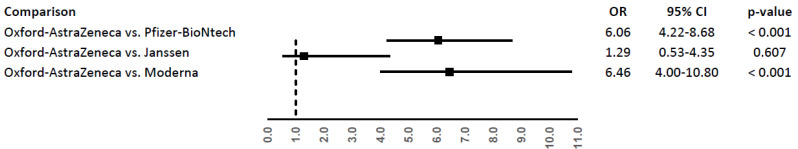
Odds ratio (OR) of thromboembolic events for Oxford–AstraZeneca vaccine compared to Pfizer-BioNtech, Moderna, or Janssen vaccines.

**Table 1 vaccines-11-01575-t001:** Doses of COVID-19 vaccines administered in Campania Region (Italy) from 27 December 2020 to 27 September 2022 and described for gender, age group, number of doses, and type of vaccine.

Variable	Level	Vaccine Doses (N = 12,692,852)	
		N	%
Gender	Female	6,509,475	51.28
	Male	6,183,377	48.72
Age	5–11 years	242,449	1.91
	12–19 years	1,086,703	8.56
	20–29 years	1,548,365	12.20
	30–39 years	1,595,933	12.57
	40–49 years	1,957,496	15.42
	50–59 years	2,232,236	17.59
	60–69 years	1,827,060	14.39
	70–79 years	1,374,019	10.83
	80–89 years	708,684	5.58
	>90 years	119,901	0.94
	Not available	6	0.00
Doses	First	4,718,844	37.18
	Second	4,239,383	33.40
	Third	3,572,382	28.14
	Fourth	162,243	1.28
Type of vaccine	Pfizer-BioNtech	8,257,218	65.05
	Moderna	3,085,673	24.31
	Oxford–AstraZeneca	1,232,608	9.71
	Janssen	116,061	0.91
	Novavax	1292	0.02

**Table 2 vaccines-11-01575-t002:** Individual Case Safety Reports (ICSRs) of Campania Region reporting a COVID-19 vaccine as suspected and at least one vascular event for the period from 27 December 2020 to 27 September 2022.

Variable	Level	Pfizer-BioNtech (N = 398)	Oxford–AstraZeneca (N = 148)	Moderna (N = 82)	Janssen (N = 13)
Number of vascular events per ICSR	Mean	3.7	3.8	4.7	2.6
Gender	Female (%)	275 (69.10)	117 (79.05)	55 (67.07)	6 (46.15)
	Male (%)	122 (30.65)	28 (18.92)	27 (32.93)	7 (53.85)
	Unknown (%)	1 (0.25)	3 (2.03)	-	-
Age	Median (IQR)	47.97 (57.35–35.54)	56.22 (65-87–45.55)	45.00 (56.87–33.03)	27.00 (39.53–18.99)
Seriousness	Serious (%)	147 (36.93)	66 (44.59)	45 (54.88)	3 (23.08)
	Not serious (%)	251 (63.07)	82 (55.41)	37 (45.12)	10 (76.92)
Doses	First dose (%)	180 (45.23)	105 (70.95)	38 (46.34)	6 (46.15)
	Second dose (%)	131 (32.91)	9 (6.08)	19 (23.17)	-
	Third dose (%)	41 (10.30)	-	11 (13.42)	-
	Not available (%)	46 (11.56)	34 (22.97)	14 (17.07)	7 (53.85)

**Table 3 vaccines-11-01575-t003:** Thromboembolic events for COVID-19 vaccines reported in Campania Region from 27 December 2020 to 27 September 2022.

Thrombotic Events (Preferred Terms of MedDRA)	Oxford–AstraZeneca Vaccine	Janssen Vaccine	Moderna Vaccine	Pfizer-BioNtech Vaccine	Total
Thrombosis	8	1	2	6	17
Venous thrombosis	3	0	1	6	10
Deep vein thrombosis	4	0	2	3	9
D-dimer of fibrin increased	5	0	2	2	9
Thrombophlebitis of the leg	4	0	0	3	7
Thrombophlebitis	1	1	1	4	7
Phlebitis	2	0	2	3	7
Infarction	1	0	0	5	6
Pulmonary embolism	1	0	4	1	6
Thrombosis of the leg	2	0	0	3	5
Thrombosis of saphenous vein	4	0	0	1	5
Ischemia	2	0	1	1	4
Deep vein thrombosis of a limb	1	1	1	1	4
Blood clot	3	0	0	0	3
Thrombus	1	0	0	2	3
Deep vein thrombosis (limbs)	1	0	0	2	3
Fibrinogen increased	2	0	1	0	3
Thromboembolism	0	1	0	1	2
Arterial thrombosis of a limb	1	0	0	1	2
Arterial occlusion, not specified	1	0	1	0	2
Deep vein thrombosis of a leg	1	0	0	1	2
Femoral deep vein thrombosis	0	0	1	1	2
Transient ischemic attack	0	0	0	1	1
D-dimer of fibrin abnormal	1	0	0	0	1
Coagulation disorder	0	0	0	1	1
Phlebitis of the arm	1	0	0	0	1
Phlebitis of a lower limb	0	0	0	1	1
Phlebothrombosis	0	0	0	1	1
Phlebothrombosis of a lower limb	0	0	0	1	1
Ischemic stroke	0	0	0	1	1
Infarction of spleen	0	0	0	1	1
Intestinal infarction	0	0	0	1	1
Myocardial infarction	0	0	0	1	1
Cerebral ischemia	1	0	0	0	1
Chronic cerebral ischemia	0	0	0	1	1
Ischemia not specified	1	0	0	0	1
Splenic ischemia	0	0	0	1	1
Micro-embolism	0	0	0	1	1
Cerebral arterial occlusion	0	0	0	1	1
Pulmonary thromboembolism	0	0	1	0	1
Venous thromboembolism	0	0	0	1	1
Deep thrombophlebitis	1	0	0	0	1
Thrombosis of the arm	1	0	0	0	1
Femoral arterial thrombosis	0	0	1	0	1
Thrombosis of the axillary vein	0	0	0	1	1
Thrombosis of varicose veins	1	0	0	0	1
Venous thrombosis (limbs)	1	0	0	0	1
Venous thrombosis of the arm	1	0	0	0	1
Deep vein thrombosis of the arm	0	0	0	1	1
Left deep vein thrombosis	0	0	1	0	1
Total	57	4	22	63	146

**Table 4 vaccines-11-01575-t004:** Reporting rates (RRs) of thromboembolic events for 10.000 doses for each COVID-19 vaccine administered in Campania Region from 27 December 2020 to 27 September 2022.

COVID-19 Vaccine	Reporting Rate (95% CI)
Oxford–AstraZeneca	4.62 (3.50–5.99)
Janssen	3.45 (0.94–8.82)
Pfizer-BioNtech	0.76 (0.59–0.98)
Moderna	0.71 (0.45–1.08)

## Data Availability

Restrictions apply to the availability of these data. Data was obtained from Campania Region and are available after permission.

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
