# Peer review of "Thromboembolic Events after COVID-19 Vaccination: An Italian Retrospective Real-World Safety Study"

_vaccines, 2023, doi:10.3390/vaccines11101575_

Round 1

Reviewer 1 Report

This is a retrospective summary of COVID-19 vaccine-related prothrombotic events. Although it is difficult to rule out the coincidence, the large population investigation still has its guide roles for future.

This paper is well-written and only some minor language check is required.

This is an observational report to show the thrombotic events recorded locally after vaccination and did not answer any scientific question.

 There are many reports on the side effects of COVID-19 vaccination. Vaccine-induced thrombosis and thrombocytopenia (VITT) has been reported in many regions. However, this is just fill a gap in Italy to give more information to the overall scenarios.

The only advantage of this paper is large cohorts with proper report and record system. Therefore the data shall be more reliable.

The argument is thrombotic events increased after vaccination and the adenovirus vaccines have more events than mRNA vaccines. For this point, it is conclusive although it is already known.

References are appropriate

The figures shall include all types of vaccines used locally.

Author Response

Dear Refereee thank you very much for your comments

We have greatly appreciated your observations.

We will modify according to your suggestions figure

Reviewer 2 Report

The authors conducted observational study that thrombotic events followed COVID-19 vaccines in Italy. the manuscript well written to the aim however comments raised and are below.

1. would you provide the data that how many number or percentage of mixing or matches vaccines as booster shots? It may enhance the immune response when different vaccine for a booster appears as good as than using same vaccine as recommendation from EU/EEA.

2. correction required, check the start date of vaccines from 27th ,2021 in the methods but 27th, 2020 in the table 1.

3. on the table 3, recommend to remove seriousness due to lack of definition of seriousness. 

4. would you like to re-organize the thrombotic events under the categories on the table 3.  such like thrombosis followed by origin ( Brain vs. legs). meaning of blood clots, thrombus, thromboembolism, thrombosis, VT and DVT etc.. are uncertain as well as similar diagnoses are overlapped in the table.

No comments

Author Response

Dear Referee

Thank you very much for your comments. We have greatly appreciated it.

according to your observation:

  1. would you provide the data that how many number or percentage of mixing or matches vaccines as booster shots? It may enhance the immune response when different vaccine for a booster appears as good as than using same vaccine as recommendation from EU/EEA.         

Booster dose scheme in Campania Region were available

2. correction required, check the start date of vaccines from 27th ,2021 in the methods but 27th, 2020 in the table 1.

Vaccine day in italy was 27th of December 2020 https://www.epicentro.iss.it/vaccini/covid-19-piano-vaccinazione

3. on the table 3, recommend to remove seriousness due to lack of definition of seriousness. 

we will modify according to your suggestions

4. would you like to re-organize the thrombotic events under the categories on the table 3.  such like thrombosis followed by origin ( Brain vs. legs). meaning of blood clots, thrombus, thromboembolism, thrombosis, VT and DVT etc.. are uncertain as well as similar diagnoses are overlapped in the table.

We will reorganize accoording to your suggestions

Reviewer 3 Report

Important paper, still a contemporary topic

Observations

- Who reported the thrombosis event? the GP? or the hospital doctor?

- it is known that culturally the Italian healthcare system is not at the forefront in terms of computer networks / information technologies. Could it represent a bias that affects under-reporting of the thrombosis event?

- Can leg pain after vaccination be considered a vascular event? Understood as a phenomenon of venous stasis?

- It is interesting to note that at the beginning vaccination was offered in Italy to doctors and teachers, just when the Astrazeneca vaccine was most used. Is there specific data on these professional categories?

- the average age of patients (50 years) with events overlaps with a higher prevalence of phlebopathy. Are there any data on the history of phlebopathy?

- Mention in the introduction the lower risk of myocarditis with Astrazeneca compared to other vaccines?

- At one point, in patients at high risk of thrombotic events, some doctors administered prophylactic thrombotic heparin associated with the vaccine. Do you have information on this strategy?

- In some papers the association between adverse event and vaccine has been defined as serendipity (Anatol J Cardiol. 2021 Jul;25(7):522-523. doi: 10.5152/AnatolJCardiol.2021.99. PMID: 34236331; PMCID: PMC8274899) . Can the same reasoning also apply to thrombotic vascular events?

- among the thrombotic events there is no mesenteric thrombosis...

- There is no significant difference in events between the two sexes. How do you explain sex bias in vaccine myocarditis?

- Don't you have mortality data?

Moderate English editing required

Author Response

Dear Referee tbhanlk you so much for your suggestions and comment:

 - Who reported the thrombosis event? the GP? or the hospital doctor?

We have addedd this part, Adverse events reports were generated by healthcare professional including GP

  • it is known that culturally the Italian healthcare system is not at the forefront in terms of computer networks / information technologies. Could it represent a bias that affects under-reporting of the thrombosis event?
  • Dear Referee, Campania Region it is well known that has one of the most modern and updated Big Data System called SINFONIA since very early 2019. We used this BigData, of which  i have been one of the coordinator for Healthcare, to collect and analyze most of the current data
  • Can leg pain after vaccination be considered a vascular event? Understood as a phenomenon of venous stasis?
  • We did not include categories that are not part of EUDRA VIGILANCE system

- It is interesting to note that at the beginning vaccination was offered in Italy to doctors and teachers, just when the Astrazeneca vaccine was most used. Is there specific data on these professional categories? Unfortunatelly we did not in clude this data in the current paper

  • the average age of patients (50 years) with events overlaps with a higher prevalence of phlebopathy. Are there any data on the history of phlebopathy?
  • The data are related only to pharacovigilance adbverse events report. on a so BIgData it would be impposibile retrospectivelly evaluated in a short time also clinical or anamnestic details
  • Mention in the introduction the lower risk of myocarditis with Astrazeneca compared to other vaccines?
  • we added this
  • At one point, in patients at high risk of thrombotic events, some doctors administered prophylactic thrombotic heparin associated with the vaccine. Do you have information on this strategy?
  • This has been evaluated on pharmacovigilance adverse events charts were are reported  concomitant medication and were withdrawn this cases
  • In some papers the association between adverse event and vaccine has been defined as serendipity (Anatol J Cardiol. 2021 Jul;25(7):522-523. doi: 10.5152/AnatolJCardiol.2021.99. PMID: 34236331; PMCID: PMC8274899) . Can the same reasoning also apply to thrombotic vascular events?
  • we consider the thrombotic events after Vaccination  as result of immune hperactivation syndrome. On this we have published several papers  troughout pandemic
  • among the thrombotic events there is no mesenteric thrombosis...
  • We had 2 splenic thrombosis reported in table for the sutdied period time
  • There is no significant difference in events between the two sexes. How do you explain sex bias in vaccine myocarditis?
  • We did not analyze in deep the different Adverse events, it was not judged as primary aim of the papers that would underline only the incidence e frequency of thromobit events according to a pharmacovigilance evaluation
  • Don't you have mortality data?
  • None of the patients at the moment of adverse events report presented exitus as severe ADR according to FDA adverse drug reaction classification

Reviewer 4 Report

General considerations: 

Great real-life study, the publication is very important! 

The English is perfect, the presentation well conducted, good to understand. 

The limitations are well described

I would propose as an on top information, if there are data available from general diagnoses in the population and if Thrombosis was more often diagnosed in 2021 than in 2019 (as in 2020 there could be a higher incidence of thromboses due to COVID itself). 

One proposed correction: 

Table 3 – You list all the names, as they were given to the Events, but there are many repetitions: 

Please, consider putting together all deep vein thrombosis, all superficial vein thrombosis (Phlebitis, etc.), all Pulmonary embolies and so have less number of single events. 

Please put the number and the 0/000 000 so that a comparison can be better performed, as the different vaccinations were applied in very different numbers. 

Author Response

Dear Referee thank you very much for your review and suggestions here our reply

I would propose as an on top information, if there are data available from general diagnoses in the population and if Thrombosis was more often diagnosed in 2021 than in 2019 (as in 2020 there could be a higher incidence of thromboses due to COVID itself). 

THe paper had the aim to analyze the whole ADR generated since vaccination till today without any distinction of time. This is related to the kind of extraction on eudra vigilance system and would be part of future studies, thanks for your suggestion

One proposed correction: 

Table 3 – You list all the names, as they were given to the Events, but there are many repetitions: Please, consider putting together all deep vein thrombosis, all superficial vein thrombosis (Phlebitis, etc.), all Pulmonary embolies and so have less number of single events. 

We have used FDA classification for eudra vigilance  ADR classifiction

Please put the number and the 0/000 000 so that a comparison can be better performed, as the different vaccinations were applied in very different numbers. 

Numbers of ADR have been inserted as unit value, thank you for suggestion